

**A Data-Efficient Deep Transfer Learning Framework for Methane Super-Emitter**
**Detection in Oil and Gas Fields Using Sentinel-2 Satellite**
Shutao Zhao[1,2], Yuzhong Zhang[2,3]*, Shuang Zhao[2,3], Xinlu Wang[1,2], Daniel J. Varon[4]
1. College of Environmental & Resource Sciences, Zhejiang University, Hangzhou, Zhejiang
Province, 310058, China
2. Key Laboratory of Coastal Environment and Resources of Zhejiang Province, School of
Engineering, Westlake University, Hangzhou, Zhejiang Province, 310024, China
3. Institute of Advanced Technology, Westlake Institute for Advanced Study, Hangzhou 310024,
Zhejiang Province, China
4. School of Engineering and Applied Sciences, Harvard University, Cambridge, United States
Corresponding Author: Yuzhong Zhang, *Email: zhangyuzhong@westlake.edu.cn;
**Abstract**
Efficiently detecting large methane point sources (super-emitters) in oil and gas fields is
crucial for informing stakeholders for mitigation actions. Satellite measurements by
multispectral instruments, such as Sentinel-2, offer global and frequent coverage. However,
methane signals retrieved from satellite multispectral images are prone to surface and
atmospheric artifacts that vary spatially and temporally, making it challenging to build a
detection algorithm that applies everywhere. Hence, laborious manual inspection is often
necessary, hindering widespread deployment of the technology. Here, we propose a novel deep-
transfer-learning-based methane plume detection framework. It consists of two components: an
adaptive artifact removal algorithm (low reflectance artifact detection, LRAD) to reduce
artifacts in methane retrievals, and a deep subdomain adaptation network (DSAN) to detect
methane plumes. To train the algorithm, we compile a dataset comprising 1627 Sentinel-2
images from 6 known methane super-emitters reported in the literatures. We evaluate the ability



of the algorithm to discover new methane sources with a suite of transfer tasks, in which training
and evaluation data come from different regions. Results show that the DSAN (average macro-
F1 score 0.86) outperforms two convolutional neural networks (CNN), MethaNet (average
macro-F1 score 0.7) and ResNet-50 (average macro-F1 score 0.77), in transfer tasks. The
transfer-learning algorithm overcomes the issue of conventional CNNs that their performance
degrades substantially in regions outside training data. We apply the algorithm trained with
known sources to an unannotated region in the Algerian Hassi Messaoud oil field and reveal 34
anomalous emission events during a one-year period, which are attributed to 3 methane super-
emitters associated with production and transmission infrastructure. These results demonstrate
the potential of our deep-transfer-learning-based method towards efficient methane super-
emitter discovery using Sentinel-2 across different oil and gas fields worldwide.

**Keywords**
Methane; Oil and gas field; Super-emitter; Sentinel-2; Deep transfer learning



## 1   Introduction


As one of the most important greenhouse gases, methane (CH4) constitutes approximately
a quarter of the overall global warming since the preindustrial age as reported by (IPCC, 2013).
Among all the sources, reducing methane emissions from anthropogenic sources, including
from oil and gas (O&G) production, is vital for mitigating near-term climate change (Lauvaux
et al. 2022). Methane emission in the O&G production sector comes from point emitters such
as malfunctioning flares, wells, storage tanks, and gas compressor stations. These point
emissions exhibit to be a long-tailed distribution, that is, a substantial fraction of the total
emissions are contributed by a limited number of anomalous point sources, which often linked
with production equipment malfunctions or abnormal operating conditions (Zavala-Araiza et
al. 2017; Duren et al. 2019). Therefore, efficiently detecting these anomalous methane point
sources is crucial for informing prompt mitigation actions.
Atmospheric methane concentrations can be quantified remotely by measuring
backscattered radiation at wavelengths (e.g., around 1700 nm and 2150 nm) that correspond to
the rotational-vibrational resonances of methane molecular transitions (Ehret et al. 2022).
Recent studies demonstrated that both multispectral and hyperspectral satellite instruments
have the capability to identify anomalous methane point emissions (Guanter et al. 2021; Varon
et al. 2021; Sánchez-García et al. 2022). Hyperspectral instruments (e.g., GHGSat, PRISMA,
EMIT, and GF-5) offer higher sensitivity to CH4 and thus lower point source detection limit
owing to their fine spectral resolution, but hyperspectral observations generally exhibit sparsity
in both spatial and temporal coverage (Naus et al. 2023; Pandey et al. 2023). In comparison,
multispectral satellites (including Landsat-8, WorldView-3, and Sentinel-2) provide global,



frequent, and spatially continuous observations, though their sensitivity to methane is lower
because of coarse spectral resolution (Varon et al. 2021; Ehret et al. 2022). As an illustration,
Sentinel-2 provides global coverage data on a weekly basis, spanning a period of eight years.
Detection limit of the Sentinel-2 measurements for methane gas in the atmosphere is roughly
5000 kg/h or greater for heterogeneous surfaces (Gorroño et al. 2023).
However, the routine scanning for methane super-emitters across varied O&G areas
remains challenging primarily due to the lack of an efficient automated source detection
algorithm (Fig. 1). Currently, source detection predominantly relies on human visual inspection,
a process that is time- and labor- consuming, thereby impeding the large-scale deployment
(Jongaramrungruang et al. 2022; Schuit et al. 2023). Deep learning techniques have been
proposed to develop point-source detectors for airborne instruments (Jongaramrungruang et al.
2022), satellite area mappers (e.g., TROPOMI) (Schuit et al. 2023), and satellite
hyper/multispectral instruments (e.g., PRISMA, Sentinel-2) (Bruno et al. 2023; Joyce et al.,
2023; Vaughan et al. 2023).
One of the key challenges in constructing such an automated detector for multispectral
observations is the low signal-to-noise ratio (SNR) in the retrieved methane signals. Because
of the coarse spectral resolution, methane signals obtained from multispectral observations are
susceptible to diverse artifacts, including interferences from vegetation, water bodies, and
smoke, making source detection a difficult task, especially over heterogenous land surface
(Cusworth et al. 2019). To mitigate these artifacts, several filtering strategies have been
proposed, such as background pixel removal (Guanter et al. 2021; Varon et al. 2021) or worst
predicted pixel removal (Ehret et al. 2022).



Another challenge arises from the necessity for an efficient detector to rapidly identify
small-scale methane point emissions in satellite data with large-scale (global) coverage.
Existing automated detectors for high-spatial-resolution satellites (Bruno et al. 2023; Joyce et
al., 2023; Vaughan et al. 2023) performed pixel-level detection which classified each pixel in
an image as plume-containing or plume-free. However, multispectral satellites such as Sentinel-
2 have high detection limits for methane emissions, even more than 5000 kg/h for
heterogeneous surfaces (Gorroño et al. 2023). This means that the retrieved images containing
methane plumes are extremely rare on both spatial and temporal scales within Sentinel-2
observations, as evidenced by Ehret et al. (2022). So far, a relatively small number of super-
emitters have been detected by multispectral satellite, mainly in desertic regions with bright,
uniform surfaces (Varon et al. 2021; Ehret et al. 2022; Irakulis-Loitxate et al. 2022; Sánchez-
García et al. 2022; Naus et al. 2023; Pandey et al. 2023). In contrast, O&G production is spread
across ~ 100 countries worldwide, often with distinct environments (EIA; https://www.eia.gov),
resulting in different noise and artifact characteristics. Therefore, an image-level detector is
required to efficiently filter out the myriad of methane-free patches. To this end, deep transfer
learning becomes a valuable strategy towards constructing a data-efficient detection model
using a limited volume of real training data (Jiang et al. 2022), without the need to construct
large simulated datasets (Jongaramrungruang et al. 2022; Radman et al. 2023). Utilizing the
inherent resemblance between the source and target domains, a deep transfer learning technique
can adapt the learned feature distribution acquired from a source data/task to a target data/task
during the training process (Iman et al. 2023).
In this work, we aim to improve methane source detection using Sentinel-2 observations.



We develop an adaptive artifact detection and masking algorithm that enhances the signal-to-
noise ratio for retrieved methane signals, and a deep transfer learning method that improves
detection efficiency and performance of discovering unknown sources, leveraging knowledge
acquired from known methane sources. To train our method, we also construct a dataset of
Sentinel-2 methane retrievals comprising Sentinel-2 detectable super-emitters reported in
literature. Our method is a step forward towards large-scale operational monitoring of methane
super-emitters by multispectral satellite instruments.


## 2   Methodology
### 2.1 Satellite data
We employ the Sentinel-2 Level 1C (L1C) top-of-atmosphere reflectance product, which
is freely available through [https://dataspace.copernicus.eu]. The Copernicus Sentinel-2
mission is composed of two polar-orbiting satellites: Sentinel-2A, launched on June 23, 2015,
and Sentinel-2B, launched on March 7, 2017. The mission can provide global coverage data
with a revisit time of 2-5 days and a swath width of 290 km. The MultiSpectral Instruments
(MSIs) onboard Sentinel-2 incorporates 13 channels spanning the visible and near-infrared
spectra, featuring spatial resolutions that vary between 10 to 60 m. Sentinel-2 data have been
used to support a variety of applications including land management, natural resource
monitoring, and risk mapping (Ienco et al. 2019; Ramoelo et al. 2015; Varghese et al. 2021).
Recent studies demonstrated the potential of Sentinel-2 to monitor methane super-emitters
(Ehret et al. 2022; Gorroño et al. 2023; Radman et al. 2023; Varon et al. 2021; Vaughan et al.



2023). Here, we use bands 11 (1610 nm) and 12 (2190 nm) for methane signal retrieval and
bands 3 (560 nm), 8 (842 nm), and 11 (1610 nm) for artifact filtering. We resample the data to
20-m resolution using the ESA snap-python toolbox and discard scenes with cloud coverage
greater than 80%.

To train our algorithm, we collect Sentinel-2 observations in the vicinity of six O&G

methane sources (indexed as #1-#6) where reoccurring ultra-emissions have been reported
(Irakulis-Loitxate et al. 2022; Sánchez-García et al. 2022; Varon et al. 2021; Zhang et al. 2022).
Table 1 summarizes the information about these methane sources, which are located in five oil
and gas fields differing substantially in surrounding terrain and surface characteristics. These
O&G sources also differ in the types of emitting facilities (e.g., compressor station, flare, well
pad, and pipeline) and the magnitude of emission fluxes (2-100 t/h) (Table 1). To construct our
training dataset, we use Sentinel-2 tile 40SBH during March 2017 to March 2023 for emitter
#1, #2, and #3, tile 32SKA from January 2019 to December 2022 for emitter #4 and #5, and tile
13SGR from January 2018 to December 2020 for emitter #6 (Table 3). We crop the original
Sentinel-2 data to generate patches of 16 km$^2$ in size, which are then used by our algorithm.

**Table 1** Reported methane super-emitters detected by multispectral satellite instruments.

| Index | Emitter [a] | Ordinates | O&G field | Land cover [b] | Country | Emission flux range (kg/h) [c] | References |
|---|---|---|---|---|---|---|---|
| #1 | Compressor station | (38.19393°, 54.19764°) | Korpeje | Barren area | Turkmenistan | 3500-92900 (08/2015-10/2020) | (Varon et al. 2021) |
| #2 | Flare | (38.33078°, 54.02832°) | Gamyshlja Gunorta | Barren area | Turkmenistan | ≥ 1800 (01/2017-11/2020) | (Irakulis-Loitxate et al. 2022) |
| #3 | Flare | (37.90825°, 53.89857°) | Keymir | Barren area and Grass land | Turkmenistan | | |
| #4 | Well-pad device | (31.6585°, 5.9053°) | Hassi Messaoud | Barren area | Algeria | 2600-29100 (10/2019-09/2020) | (Varon et al. 2021) |
| #5 [d] | Pipeline | (31.778°, 5.995°) (31.768°, 6.000°) | Hassi Messaoud | Barren area | Algeria | 3100 (12/29/2020) 2500 (12/29/2020) | (Sánchez-García et al. 2022) |



| #6 | Compress or station | (31.7335°, -102.0421°) | Permian basin | Shurbland | U.S. | 2360-21830 (07/2020-09/2020) | (Zhang et al. 2022) |
|---|---|---|---|---|---|---|---|

[a] Reports of these sources are all based on Sentinel-2 data expect for #5 which is based on Worldview-3.
[b] Land cover type near the emitter is obtained from the annual ESA/CCI land cover map 2020
[https://maps.elie.ucl.ac.be/CCI/viewer/index.php] as a reference. It is noted that the land cover map has a spatial
resolution of 300 m, which cannot reflect surface features smaller than an area of 300 m$^2$.
[c] Values in this column represent emission flux during the time range or date studied in literatures. It is noted that the
emission flux of emitter #2-3 has not been reported by (Irakulis-Loitxate et al. 2022), and 1800 kg/h is the detection
limit of Sentinel-2 provided in the literature.
[d] Emitter #5 contains two pipeline leakage sources approximately 1.2 km apart. They are numbered together since
they are only around 60 pixels apart in the 20m resolution Sentinel-2 image.
**2.2 Framework for multispectral satellite point source detection and quantification**
Fig. 1 shows the workflow of methane super-emitter monitoring using Sentinel-2 satellite
data, with algorithms developed in this study highlighted in red text. The workflow primarily
includes three steps, methane signal retrieval, source detection, and flux quantification.
First, methane signals are retrieved from satellite measurements. We employ the structural
similarity index measure (SSIM) algorithm (Zhou et al. 2004) to filter out cloudy observations
and the low-reflectance adaptive detection (LRAD) algorithm developed in this study (Section
2.3) to filter out other interference. We then compute fractional methane absorption signal (ΔR,
unitless) using band 11 and 12 from Sentinel-2 (Ehret et al. 2022; Irakulis-Loitxate et al. 2022):
$$\Delta R^t = \frac{band_{12}^t / band_{12}^{ref}}{band_{11}^t / band_{11}^{ref}}$$

where $band_{12}^t$ and $band_{11}^t$ represent observations on the date of interest (t) and $band_{12}^{ref}$ and
$band_{11}^{ref}$ represent reference conditions without any methane enhancement. We borrow the idea
of sliding time window in Ehret et al. (2022) to predict $band_{12}^{ref}$ and $band_{11}^{ref}$ by the multivariate
linear regression (MLR) model trained on band 11 and 12 observations in the time window
(within 60 days prior to date t). Data excluded by SSIM and LRAD are not used for the MLR
model training. See Text S1 for detailed information on the methane signal retrieval step.
Second, we train an automated detector to detect potential methane super-emitters based on



retrieved ΔR, in place of human inspection. We annotate ΔR images retrieved from Sentinel-2
observations of 6 methane super-emitters (Table 1). The dataset is then used to train and
evaluate a deep subdomain adaptation network (DSAN) (Section 2.4) to detect whether an
image contains methane plumes. Our work demonstrates that the DSAN detector, trained with
a relatively small number of annotated ΔR images, shows promising performance in unknown
source detection.

Finally, we quantify emission fluxes (kg/h) of detected methane plumes by employing the

Integrated Mass Enhancement (IME) method (Frankenberg et al. 2016; Varon et al. 2018). See
Text S2 for detailed descriptions about the flux quantification method.

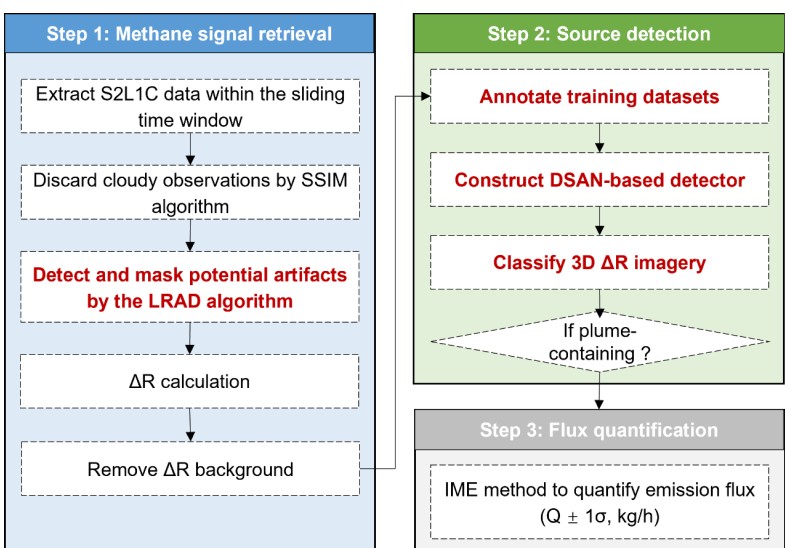


**Fig. 1.** The methane super-emitter monitoring workflow (from Sentinel-2 L1C product to emission
flux of the detected methane point emission signal). Text in red highlights the novel algorithms
developed in this study.
**2.3 Low reflectance artifact detection (LRAD) algorithm for artifact removal**

To increase the signal-to-noise ratio of Sentinel-2 methane retrieval, we develop a low

reflectance artifacts detection (LRAD) algorithm to identify and remove varied artifacts



associated with low reflectance in the methane-sensitive band by surface features. Figure 2 (a)
and (b) show examples of these potential artifacts resulting from varied surface elements
including smoke (from burning flare), rocky soil (with high mineral content), dark soil (with
high organic matter or water content), water body, cloud shadow, and vegetation (Gorroño et
al. 2023; Naus et al. 2023). These artifacts in the SWIR bands may be filtered out by leveraging
additional bands that are sensitive to the artifacts but insensitive to methane (Figure 2(c)).
Fig. 3 shows the pseudocode of the LRAD algorithm, which creates a surface artifact mask
using Band 3 (560 nm), 4 (665 nm), and 8 (842 nm), in addition to Band 11 and 12. For
combustion-related artifacts, the algorithm first filters out pixels with saturated reflectance in
Band 11 and 12, which are related to thermal anomalies from high-temperature combustion
(Liu et al. 2021). The algorithm then filters out pixels affected by heavy smoke, identifiable by
extraordinarily low visible-band reflectance in Band 3 (the 5% lowest values of the scene). We
calculate the standard deviation σ and then apply the 2σ (around 95% confidence interval) as
the masking threshold. The above mask is then dilated to ensure that interference from
combustion sources is removed.
Additionally, the LRAD algorithm filters out pixels with concurrent negative values of the
Normalized Difference Vegetation Index (NDVI) (Band 8 and Band 4) and the Normalized
Difference Built-up Index (NDBI) (Band 8 and Band 11), which are related to low-reflectance
objects in SWIR such as water bodies (Biermann et al. 2020; Fan et al. 2020; Purio et al. 2022).
Positive values of these indices have been used in literature to detect healthy vegetation and
urban areas (Kuc and Chormański 2019).



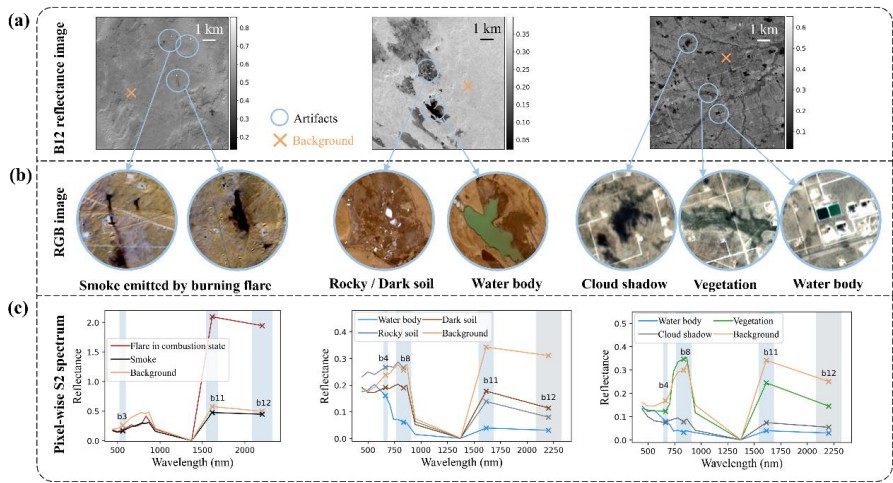


**Fig. 2.** Examples of varied artifacts in Sentinel-2 (S2) L1C reflectance images. (a) S2L1C band 12 (b12) reflectance images in Hassi Messaoud (20190117T32SKA), Gamyshlja Gunorta (20200404T40SBH), and Permian basin (20190126T13SGR). (b) Representative RGB images of the artifacts presenting low reflectance in b12. (c) Pixel-wise S2L1C reflectance spectrum of the background and representative artifacts. Bands used for identifying artifacts are shown in blue shadings.

---

**Algorithm** Low reflectance artifacts detecting (LRAD) algorithm

**Input:** Data cube $X$ with size of $m \times n \times 5$ is extracted from S2L1C product, each pixel $i$ in $X$ has 5 wavelength bands including $b_3$, $b_4$, $b_8$, $b_{11}$, and $b_{12}$.

**Output:** $Mask$

1:   Initialize $Mask = Ones(m \times n)$
2:   **for all** $i$ **do**
3:      **if** $(b_{11}^i \geq 1.0 \, \& \, b_{12}^i \geq 1.0)$ **then**          //Detect flare in combustion state
4:         $Mask[i] = 0$                     //Filter pixels containing flare
5:         $Mask[where \, (b_3 \leq Quantile_{b_3}^{5\%})] = 0$   //Filter pixels containing smoke
6:      **end if**
7:      $NDVI = (b_8 - b_4)/(b_8 + b_4)$; $NDBI = (b_{11} - b_8)/(b_{11} + b_8)$
8:      $Mask[where \, (NDVI \leq 0) \cup where \, (NDBI \leq 0)] = 0$
         //Filter pixels containing artifacts with low reflectance in NIR and SWIR bands
9:   **end for**
10:  $Mask = Dilation \, (Mask)$
11:  **return** $Mask$

---

**Fig. 3.** LRAD algorithm to generate the mask for low reflectance artifacts in methane retrieval bands (Band 11 and 12) using data in Band 3, 4, and 8.

**2.4 Deep transfer learning for methane source detection**

We employ the deep subdomain adaptation network (DSAN) (Zhu et al. 2021) to detect the presence of methane plumes in retrieved ΔR images (Fig. 4). DSAN is a transfer learning



algorithm that leverages feature representations acquired from a labeled source domain to
enhance performance on the unlabeled target domain (Pan and Yang 2010). By using DSAN,
we attempt to address the challenge that a methane-source classifier trained with labeled data
in one location (source domain) tends to perform inadequately in another location where labeled
data are unavailable (target domain), because of great differences in surface characteristics
between regions (domain shift).

Fig. 4 illustrates the structure of DSAN applied in this study. DSAN consists of deep

feature extraction blocks and a domain adaptation module. Feature extraction is done by
adapting a pre-trained residual neural network (ResNet-50) as the backbone of DSAN. ResNet-
50 has demonstrated exceptional performance in various image classification tasks, especially
those based on spatial context, largely because of its strong feature mining capability enabled
by shortcut connections (Burke et al. 2021) (see Fig. S2). ResNet-50 consists of 16 residual
blocks that contain a series of convolutional layers and shortcut connections. Following each
convolutional layer, there is a subsequent batch normalization layer and a Rectified Linear Unit
(ReLU) activation function.

The domain adaptation module transforms deep features extracted by ResNet-50 to align

the feature distributions between source and target domains. The alignment is performed based
on local maximum mean discrepancy (LMMD), which measures the distance between feature
distributions (Zhu et al. 2021). The general form of LMMD is presented as:
$$LMMD(P, Q) = \frac{1}{N} \sum_{i=1}^{N} \left\| E_P^i \left[ \phi(D_s^i) \right] - E_Q^i \left[ \phi(D_t^i) \right] \right\|_H^2$$
Where $D_s$ and $D_t$ are the samples in source and target domain, $P$ and $Q$ are the probability
distribution of $D_s$ and $D_t$, and $i$ is the class of the sample (plume-containing or plume-free).



LMMD is designed to capture both global (whole dataset) and local (each class) domain
differences, and therefore is sensitive to variability within each class. This property is important
for our application because the difference between the two classes (plume-containing and
plume-free ΔR images) are more subtle compared to a typical image classification task.

The DSAN is first trained using labelled ΔR images in the source domain and unlabeled

ΔR images in the target domain, before it is used to predict labels for target-domain images.
The input ΔR imagery is transformed to match the ResNet-50 (which serves as the backbone of
DSAN) input format. Before feeding into the network (Fig. 4), the input image was resized to
224*224, augmented by randomly flipping the images horizontally during the training process,
and then normalized to ensure that the three channels had a consistent scale. The model is
trained with a learning rate of 0.001 using stochastic gradient descent (SGD) optimizer over
100 epochs.

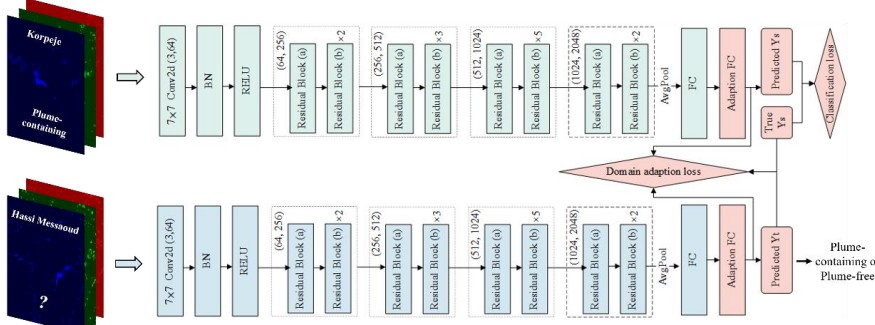


**Fig. 4.** The architecture of DSAN. DSAN employs ResNet-50 to learn features from labeled (green)
and unlabeled (blue) data, and then the domain adaptation module (red) to reduce the domain
distribution discrepancy.
**2.5 Experiment design**
**2.5.1    Performance evaluation on transfer tasks**

We design two experiments (Fig. S4) to evaluate the performance of the DSAN framework



in detecting unknown sources, using 6 ΔR datasets corresponding to the 6 super-emitters
(denoted as #1-6; Table 1) for training and evaluation. Table 2 describes the training, validation,
and test subsets separation ways. In the first experiment ('1→1' task), we use one of the six
datasets as the source domain (labels available to the algorithm) and another dataset as the target
domain (labels unavailable to the algorithm and to be predicted). In total, there are 6×5=30
'1→1' tasks to be evaluated. In the second experiment ('5→1' task), we use five of the six
datasets as the source domain and the remaining one as the target domain, which yields six
'5→1' tasks. The '1→1' tasks examine how well a detector constructed based on data from a
known source can discover unknown sources, while the '5→1' tasks evaluate whether and to
what degree performance can be enhanced by including training data from multiple sources.

To compare, we also build two convolutional neural networks (CNNs) (Fig. S3) based on

MethaNet (Jongaramrungruang et al. 2022) and ResNet-50, which, unlike DSAN, do not
contain a domain adaptation module. For each '1→1' or '5→1' task, a CNN methane-source
detector is trained with the labeled source-domain dataset(s) before being applied to predict the
labels for the target domain. We train the MethaNet model from scratch and the ResNet-50
model with a fine-tuning strategy demonstrated by (Radman et al. 2023).
**Table 2** Training, validation, and test subsets separation for different types of models and tasks.

| Model | Task | Training set | Validation set | Test set |
|---|---|---|---|---|
| DSAN | '1→1', '5→1' | source domain | --- | target domain |
| MethaNet and ResNet-50 | '1→1', '5→1' | 80% source domain | 20% source domain | target domain |
|  | non-transfer | 80% source domain | 20% source domain | --- |

The performance is assessed for each task with accuracy, precision, recall, and the macro-

F1-score using the scikit-learn package (Pedregosa et al. 2011). The main metric we use is the
macro-F1 score, computed as the average of F1 scores for each class (harmonic mean of
precision and recall). The macro-F1 score has a range of 0-1, suitable for datasets with



imbalanced positive and negative samples. A higher macro-F1 score indicates a better overall
performance. Additional metrics encompass accuracy, representing the ratio of correctly
predicted instances to the total instances; precision, calculated as the number of true positive
predictions divided by the total number of positive predictions; and recall, determined by
dividing the number of true positive predictions by the total number of actual positive instances.
**2.5.2    Real-world application for new source discovery**
To test in a real-world scenario, we apply the proposed workflow (Fig. 1) to the Hassi
Messaoud O&G field in Algeria. We randomly select an orbit (for tile T32SKA) in this region
which covers an area of $4\times108$ km$^2$ during July 2019-June 2020. The original data are
segmented and converted into 200px × 200px patches (an area of ~16 km$^2$), generating a total
of 3537 cloud-free ΔR images in the region. We use these unannotated data as the target domain
for DSAN and the labeled datasets described above (#1-#6) as the source domain. Finally, the
results predicted by the detector are evaluated against manually determined labels.
**3.   Methane retrieval (ΔR) imagery dataset**
We compile ΔR datasets containing six super-emitters reported in the literatures (Table 1)
using Sentinel-2 L1C observations. Each sample in the dataset consists of a ΔR image retrieved
from the original satellite data (Step 1 in Fig.1) and a label determined manually indicating the
presence or absence of methane sources (plume-containing or plume-free).
The ΔR images of the dataset are processed with the LRAD algorithm (Section 2.4). Fig. 5
shows examples of artifact masks generated by LRAD and compares the ΔR images with and
without applying the masks. This result demonstrates that the algorithm can detect and remove
varied types of surface artifacts, including dark soil, rocky soil, water body, burning flare,



smoke plume, vegetation, and cloud shadow. Fig. S6 presents additional examples that LRAD
generates masks that are adaptive to temporal changes in land covers, thus capable of detecting
seasonally varying artifacts. As shown in Fig. 5, removing of these artifacts by the LRAD
algorithm enhances signal-to-noise ratios (SNRs) (defined as $SNR = 20 * \log_{10}(avg./std.)$,
$avg.$ and $std.$ are calculated from the entire $\Delta$R image) in $\Delta$R images by 12.12-42.30%,
facilitating the following source detection step. Fig. S7 compares the averaged SNRs of the six
$\Delta$R datasets before and after deploying the LRAD algorithm.

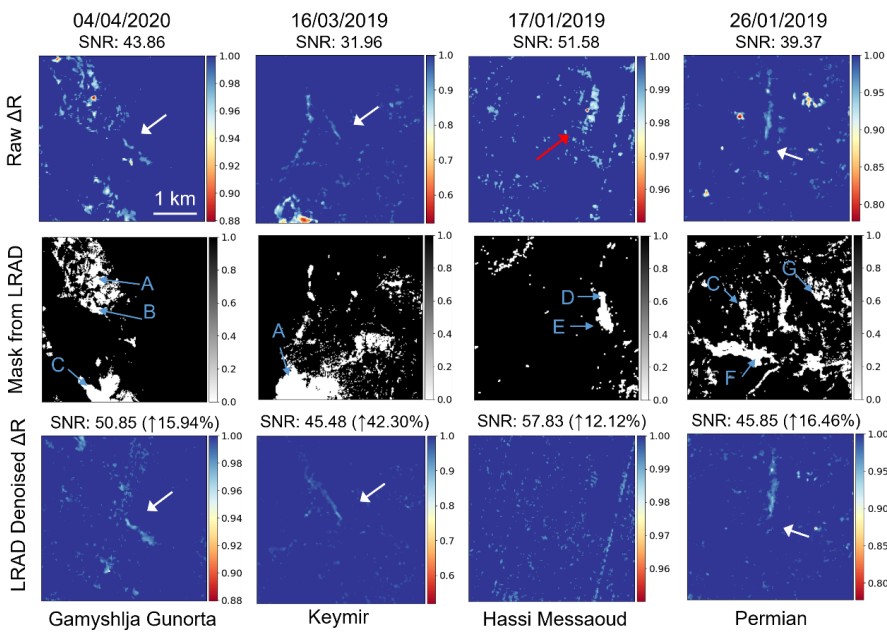


**Fig. 5.** Examples of the $\Delta$R images and masks. The first row showed the raw $\Delta$R images outputted
by Step 1 procedures (Fig. 1) without LRAD deployed, the second row displayed the latent artifacts
masks generated by LRAD algorithm, and the third row exhibited the denoised $\Delta$R images outputted
by Step 1 procedures (Fig. 1) with the LRAD performed. White arrows indicated true methane
plumes, and red arrow indicated plume-like artifacts. Blue characters and arrows in the binary masks
pointed to different types of the latent artifacts.
We label the $\Delta$R image following the decision rule as described in Fig. 6 and Text S3. Table
3 summarizes the information of the methane imagery dataset retrieved from Sentinel-2 L1C



data. The dataset consists of subsets of 6 super-emitters reported in the literature (Table 1). Each
subset contains 200-400 samples. These subsets differ greatly in the ratio between positive
(plume-containing) and negative (plume-free) samples, ranging from 8.1% in #6 to 81.95% in
#1, reflecting large variations in emission frequencies among varied sources. Most of the
positive samples contain one methane plume, except for #5 in which occasionally two methane
plumes are present simultaneously. We quantify the emission rates of positive samples using
the IME method (Text S2) (Fig. S5). The average emission flux varies from 1952 kg/h in #5 to
17122 kg/h in #3. Moreover, the background noises exhibit considerable variations among the
six subsets (Fig. 7). Subsets #1, #4, and #5 present uniform noises originating from
homogeneous surfaces yet subsets #2, #3, and #6 have greater heterogeneity resulting in a
higher occurrence of artifacts.

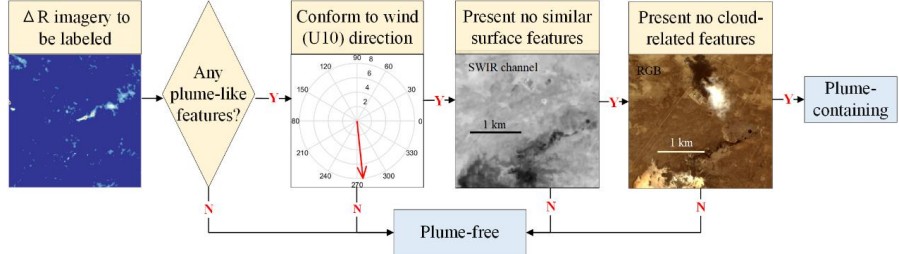


**Fig. 6.** A flowchart of the labeling decision rule of ΔR imagery (Detailed description is provided in
Text S3).

**Table 3** Description of the six labelled ΔR datasets.

| Index | Sentinel-2 tile ID | Time span | Number of plume-containing observations | Number of plume-free observations | Average emission flux (kg/h) |
|---|---|---|---|---|---|
| #1 | | | 109 | 133 | 11076 |
| #2 | T40SBH | 03/2017-03/2023 | 95 | 164 | 8826 |
| #3 | | | 66 | 186 | 17122 |
| #4 | T32SKA | 01/2019-12/2022 | 92 | 233 | 5717 |
| #5 | | | 128 | 181 | 1952 |
| #6 | T13SGR | 01/2018-12/2020 | 18 | 222 | 14443 |

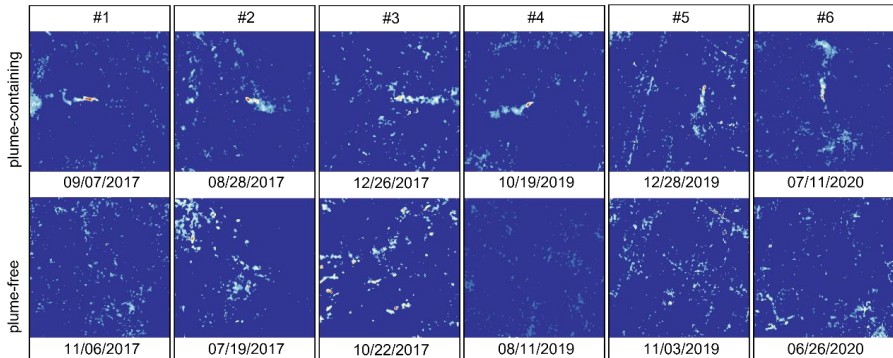


**Fig. 7.** Examples of the plume-containing and plume-free images in ΔR datasets #1-#6.
**4. Performance evaluation of the DSAN model**
Fig. 8 evaluates the ability of the DSAN model to detect a methane source in an unannotated
region (transferability) with the macro-F1 scores achieved for varied '1→1' or '5→1' transfer
tasks (Section 2.5.1). To compare with conventional CNNs, Fig. 9 shows results of MethaNet
and ResNet-50 for the same tasks. In addition to macro-F1 scores, Table S1-S3 also tabulate
other performance metrics from the experiments including accuracy, precision, and recall.
The DSAN model achieves average macro-F1 scores of 0.86 (0.69 to 0.93) for the '1→1'
tasks and 0.89 (0.77 to 0.94) for the '5→1' tasks (Fig. 8), which consistently outperforms both
MethaNet (0.70 for '1→1' tasks and 0.76 for '5→1' tasks) (Fig. 9(a)) and ResNet-50 (0.77 for
'1→1' tasks and 0.81 for '5→1' tasks) (Fig. 9(b)). The performance of conventional CNN
models degrades substantially in these transfer tasks (off-diagonal of Fig. 9), compared to non-
transfer tasks (training and validation data from the same locations) (average macro-F1 scores
are 0.87 for MethaNet and 0.95 for ResNet-50) (diagonal of Fig. 9), demonstrating the
challenges of transfer tasks. Moreover, the performance of CNNs in '5→1' tasks (rightmost
column of Fig. 9), only marginally improved over their performance in '1→1' tasks (left six
columns of Fig. 9), is still inferior to DSAN's performance in most '1→1' tasks (Fig. 8), which



indicates that including a limited number of training samples from diverse regions is insufficient
for conventional CNNs to enhance their transferability, underscoring the value of the transfer
learning algorithm such as DSAN.

The disparity of the performance presented above can be interpreted by comparing the deep

features extracted by MethaNet, ResNet-50, and DSAN. Fig. 10 maps high-dimensional deep
features to a 2-dimentional plot generated by the t-distributed stochastic neighbor embedding
(t-SNE) algorithm (Laurens van der Maaten and Hinton 2008). Blue points are source domain
samples and orange points are target domain samples. DSAN exhibits better alignment between
the source and the target domains compared to MethaNet and ResNet-50. In the DSAN
subfigures, it is evident that not only are the source and target points well-aligned, but samples
belonging to different classes also exhibit noticeable distinctions. This result is consistent with
our understanding that the domain transfer module in the DSAN model can effectively close
background differences between different regions (domain shift), enhancing the ability of the
algorithm to identify methane plumes at a new location.

Fig. 8 and Fig. 9 also indicates that some of the datasets appear more difficult to predict

than others. The DSAN's performance for dataset #2 and #6 is not as good as for other datasets
(Fig. 8), while MethaNet performs poorly for dataset #2, #5, and #6 and ResNet-50 performs
poorly for dataset #2 and #6 (Fig. 9). Some dataset characteristics may have contributed to
lower performance. Dataset #2 is marked by highly heterogeneous surface, Dataset #5 by
smaller methane fluxes and plume sizes, and Dataset #6 by higher surface complexity and
imbalanced positive / negative classes (Fig. 7 and Table 3).

Increasing the source domain from one dataset ('1→1' tasks) to five ('5→1' tasks) slightly





improves the performance of the DSAN model (Fig. 8), demonstrating the benefit of including
more and diverse training samples. However, #6 remains the most difficult dataset with no
improvement.

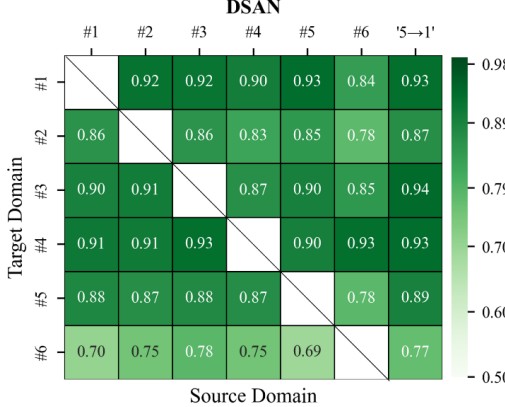


**Fig. 8.** Macro-F1 scores on the transfer tasks given by DSAN. Each square represents a transfer task.
'5→1' represents the source domain is fused by five datasets except for the target domain dataset.

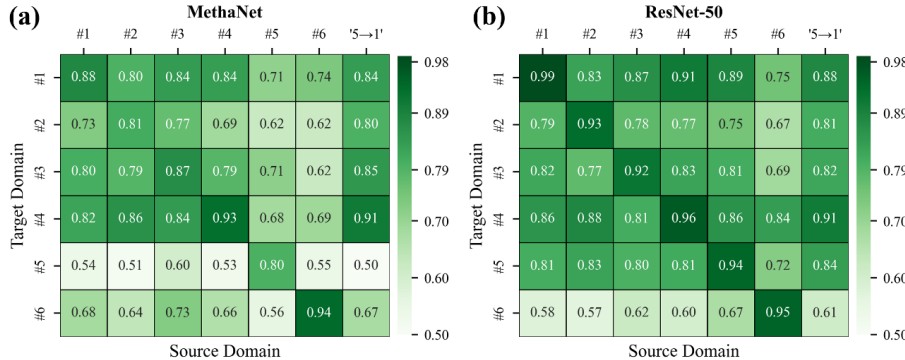


**Fig. 9.** Macro-F1-scores given by (a) MethaNet and (b) ResNet-50. Each square represents a task.
Tasks on the diagonal pertain to non-transfer tasks, with each dataset partitioned into a training set
(80%) and a validation set (20%). Tasks outside the diagonal are transfer tasks. '5→1' denotes that
the source domain is fused by five datasets except for the target domain dataset.

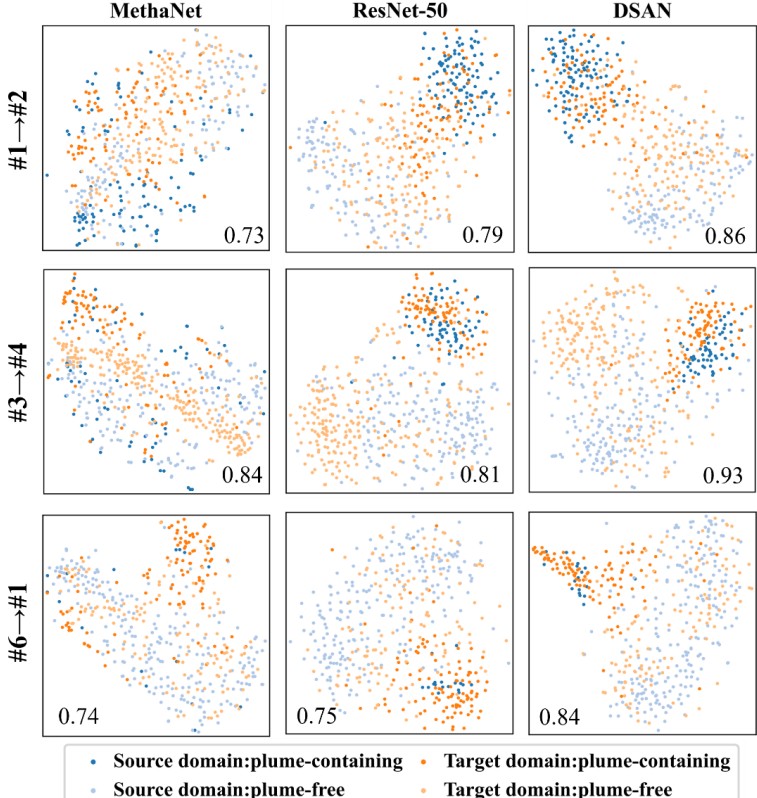

**Fig. 10.** t-SNE visualizations of the learned feature representations of the ΔR datasets across different models and transfer tasks, providing insights into domain shift and how well the well-trained models identify different classes in the target domain. From the left to right column: MethaNet, ResNet-50, and DSAN on three '1→1' transfer tasks (#1→#2, #3→#4, and #6→#1). Each point represents a data sample. The number in each subfigure denotes the macro-F1 score of the target domain label predicted by the model.

## 5. Real-world application for methane source discovery

We apply the proposed AI-assisted monitoring workflow (Fig. 1), including the LRAD and DSAN algorithms, to a 432 km$^2$ area (Fig. 11) in the Hassi Messaoud O&G field in Algeria (Section 2.5.2). The algorithm processed in total 3527 images (200 pixel by 200 pixel) for one year, yielding 3168 negative (plume-free) and 369 positive (plume-containing) detections.

We manually verified that 33 out of the 369 positive detections contain true methane plumes from three methane super-emitters (denoted as P(1), P(2), and P(3) in Fig. 11) and that 1 false



negative detection was identified at P(2) (see Fig. S8). Using the Google Earth Map, we
attributed P(1) to a production well (31.8651°N, 6.1683°E) and P(2) to pipeline leakage
(31.7566°N, 6.1864°E). We did not identify OG infrastructure associated with P(3)
(31.5846°N, 6.4878°E) from the Google Earth Map. Fig. 11 presents visual imagery of each
source and the true positive plumes detected by our method. These super-emitters were not
known at the time of our experiment. Two recent studies reported P(1) based also on Sentinel-
2 data (Naus et al. 2023; Pandey et al., 2023).
Methane plumes are detected twice at P(1), 30 times at P(2), and twice at P(3) during July
2019 to June 2020 (Fig. 12), resulting in respective detection frequencies of 1.6%, 24%, and
1.6% for the three sources after cloudy days are excluded. Meanwhile, the LRAD algorithm
detects flaring as a byproduct (Fig. S9). We detected 67 flaring events at P(1) and one flaring
event at P(2) (Fig. 12). Flaring detection at P(1) occurs primarily during July to August 2019
and January to May 2020.
We quantified the emission fluxes of the three sources using the IME method (Varon et al.
2021) (see Text S2 for details about the method). The average emission rate is 31133 kg h$^{-1}$
for P(1), 3990 kg h$^{-1}$ for P(2), and 8210 kg h$^{-1}$ for P(3) (Fig. 12). The largest emissions were
found at P(1) due to a blowout event with 18421±6575 kg h$^{-1}$ on January 4, 2020 and 43845
±9169 kg h$^{-1}$ on January 7, 2020. This result is generally comparable to estimates given by
Pandey et al. (2023) (21000±6000 kg h$^{-1}$ on January 4) and Naus et al. (2023) (29800±14900
kg h$^{-1}$ on January 4 and 68400±34200 kg h$^{-1}$ on January 7).

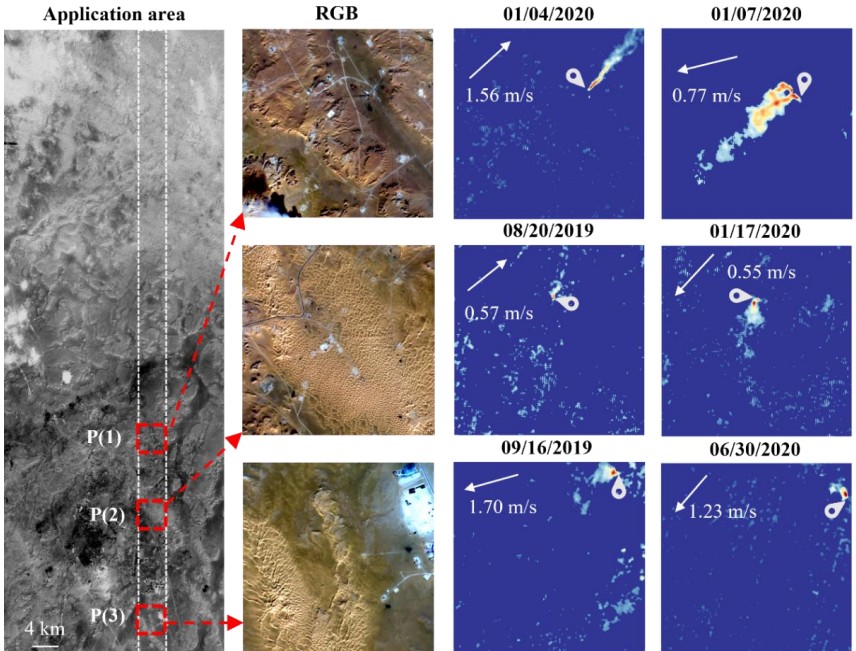

P(1): 31.8651°, 6.1683° | P(2): 31.7566°, 6.1864° | P(3): 31.5846°, 6.4878°

**Fig. 11.** From left to right: Application area (the rectangular area within the white dotted line) extracted from Sentinel-2 data, RGB images of the positive patches containing methane point sources (P(1)-P(3)), and examples of the methane plume-containing ΔR images detected by our method. The white pin in ΔR image points to the source location.

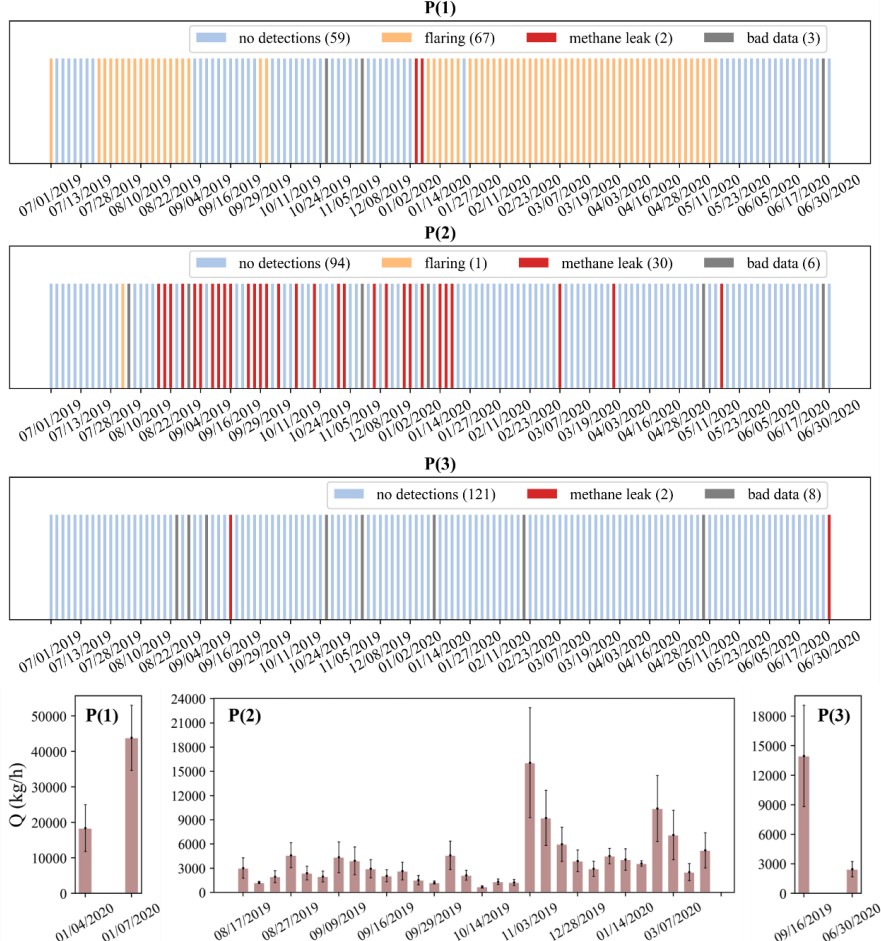

**Fig. 12.** Time series of the detected methane leaking events, flaring, and the retrieved emission flux of the methane plumes for P(1), P(2) and P(3). It is noted that detected methane leaks and flaring come from different facilities, and the flare burn dates do not coincide with the leak dates. No detections indicate methane-free and flaring-free. Bad data mainly indicates cloudy data or data that is fully covered by artifacts.

Table 4 summarizes the performance metrics for the real-world application. Our algorithm demonstrates a good detection capability with an accuracy of 0.90, consistent with the averaged value for the 36 transfer tasks (section 4.1.1). This performance surpasses the detection accuracy of approximately 0.80 reported by the CH4Net which used Sentinel-2 for the west coast of Turkmenistan (Vaughan et al. 2024). For 3168 plume-free images, the DSAN detector





achieves a false positive rate of 0.096 (FP / TN+FP), higher than the results of existing detectors
tested on synthetic datasets (Zortea et al. 2023; Rouet-Leduc and Hulbert 2024). Nonetheless,
this rate is lower than the 0.14 reported by the U-Plume detector on GHGSat-C1 observations
(Bruno et al. 2023) and the 0.18 reported by (Vaughan et al. 2024).

Additionally, our detector shows the macro-F1 score of 0.56, which is lower than that

reported in Section 4 for the evaluation tasks primarily due to the 336 false positive detections.
Further analyses suggest that these false positives are related to smoke, built-up, land surface,
and cloud/cloud-shadow (Fig. 13(a)). We categorize these false positives based on the type of
main artifacts (Fig. 13(b)). Artifacts related to land-surface variability accounts for 77.61% of
the false positives, followed by those related to cloud or cloud shadow (19.10%), and smoke
(3.28%). These results indicate that some artifacts remain after processed by the artifact-
removal algorithm LRAD. Investigation into these artifacts, particularly those by land surfaces,
is key to further improving the performance.
**Table 4** Manual validation of detections by the AI-assisted framework.

| 07/2019 - 06/2020 | TP[a] | FP[b] | TN[c] | FN[d] | Precision | Recall | Macro-F1 score | Accuracy |
|---|---|---|---|---|---|---|---|---|
| All 3537 patches of the swath | 33 | 336 | 3167 | 1 | 0.09 1.00 | 0.97 0.90 | 0.56 | 0.90 |

[a-d] TP (true positive), FP (false positive), TN (true negative), and FN (false negative) represent specific
categories of predictions



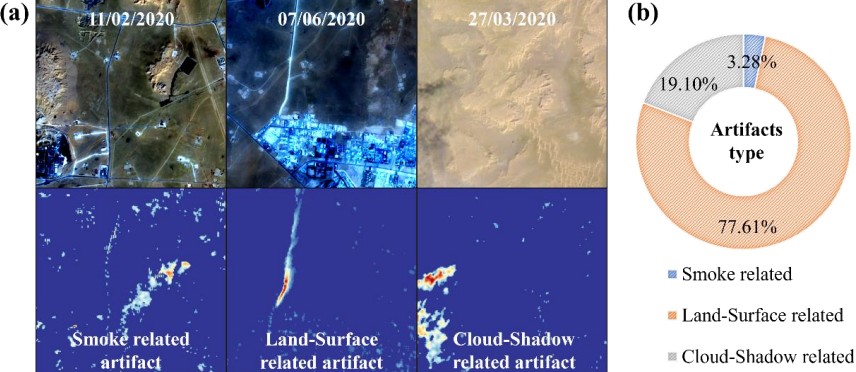

**Fig. 13.** False positive detection in the real-world application. (a) Representative examples of the false positive results, and the corresponding RGB images extracted from Sentinel-2 L1C product; (b) Contributions of various artifact types to false positive detections.

## 6. Discussion

### 6.1 Comparison with existing denoising methods

Noise and artefacts in retrieved ΔR imagery poses significant challenges to real-world image classification tasks such as satellite-based methane plume detection, impacting the convergence and generalization of deep neural networks (Dodge and Karam 2016). Table 5 summarizes existing denoising methods. To reduce noises, Varon et al. (2021) proposed to remove outliers using 3×3 median filter algorithm and remove background noises below 95% confidence interval. Similarly, Ehret et al. (2022) discarded the 5% worst predicted pixels obtained from methane-free background estimation and then apply a Gaussian filter. Furthermore, Zortea et al. (2023) generated a binary mask to exclude the water-body-related artifacts using the MNDWI. These denoising methods performed well on relatively homogeneous surfaces, where noise is uniformly distributed and artefacts are small in area and infrequent in time. However, in heterogeneous regions, such as those shown in the first and last columns of Fig. 5, artifacts are more prominent and often cover areas larger than those of the



methane plumes, making them more challenging to address by existing denoising methods.
Utilizing additional spectral bands, our LRAD algorithm is designed to address multiple types
of artifacts and is adaptive to different types of land surfaces. As illustrated in Fig. S6, LRAD
generates large-area denoising masks for heterogeneous surfaces and small-area or even no
masks for homogeneous regions. The effectiveness of this approach is further demonstrated by
the SNR improvements shown in Fig. S7.
**Table 5** Summary of existing denoising methods.

| References | Denoising method | Used Sentinel-2 band |
|---|---|---|
| Varon et al. (2021) | 3×3 median filter & background mask: [methane enhancement > 95[th] percentile] | --- |
| Ehret et al. (2022) | Gaussian filter & 5% worst prediction pixels | b11, b12 |
| Zortea et al. (2023) | Gaussian filter & water body mask: [MNDWI >0.2] | b3, b12 |
| This study | LRAD | b4, b8, b11, b12 |


**6.2 Comparison with existing methane detectors**
Multispectral satellite instruments such as Sentinel-2 record high-spatial-resolution global
data, potentially capturing methane plume signals from numerous super-emitters. It poses great
challenges to detect methane plumes from vast areas with various background noises relying
on visual inspection, as well as to extensively annotate real-world training data for constructing
automated detectors. Recently, various deep learning architectures have demonstrated
feasibility for the automated detection of methane super-emissions in satellite imagery,
including the vision transformer based network (Rouet-Leduc and Hulbert 2024), U-Net based
models (Bruno et al., 2023; Vaughan et al., 2024), ResNet-50 (Zortea et al., 2023), EfficientNet-
V2L (Radman et al., 2023), and MethaNet (Jongaramrungruang et al., 2022), a specialized
network for methane detection. Most existing detectors require huge-volume simulated or
synthetic datasets, the size of which is more than 100 times larger than the real data used to



train our methane detector.

While works in a data-efficient manner, our transferable DSAN method demonstrates

lower false positive rates than existing detectors also trained with real data (Section 5). These
detectors possibly degrade performances on test sets, due to the potential domain shift arising
from spatiotemporal variations in real environment (Fig.10). In contrast, the specialized domain
adaptation architecture in our detector can bridge such domain shift, making it promising for
cost-effective and large-scale methane super-emitters detection. Once $\Delta$R imagery with labeled
information from one methane point source is available, the DSAN model can learn the
plume/noise feature representation and transfer to other geographic regions with similar or even
different environmental conditions.

**6.3 Limitations and future enhancements**

It should be noted that while the LRAD algorithm could effectively remove most artifacts

presenting low reflectance values in methane-sensitive bands, but its robustness to remove
plume-like artifacts in complex situations (see Fig. 13(a)) needs to be improved in future studies.
Our real-world application in Hassi Messaoud reported a relatively high number of 336 false
positive out of 3527 classifications. Most false positives were caused by artifacts that spectrally
overlapped with methane absorption. This result suggests that more work is needed to eliminate
these artifacts, especially those originating from surface features (account for 77.61%), to
reduce the false positive rate of the Sentinel-2 monitoring workflow. Considering that land-
surface type artifacts (from built-up areas and natural low-reflectivity surfaces) are spatially
invariant, hyperspectral or radar satellite observations can be used to pre-identify potential



artifacts in oil and gas fields. Both of them excel at discriminating among various built-up
structures and materials properties (Kuras et al. 2021). For key oil and gas fields with high
emission frequencies, an artifact library can even be constructed so that Sentinel-2 can directly
look up for regional masking when detecting methane sources. Furthermore, applications in
more O&G fields would be needed for methane ultra-emitter monitoring. Augmentation of the
true and diverse methane plume datasets can lead to better generalization capabilities of the
detection model, while the time and labor costs of annotating plume-containing images need to
be considered.

**7. Conclusions**

Here, we proposed a novel deep-transfer-learning-based approach that combined an

adaptive artifacts removal algorithm (LRAD) with a transferable plume detector (DSAN), to
identify methane-plume-containing images retrieved from Sentinel-2 observations. Our
evaluation demonstrated that the proposed method efficiently detects plumes in different O&G
fields. Applying the method to the Hassi Messaoud O&G field over a 1-year period discovered
33 anomalous emission events from three methane super-emitters, which were attributed to well
blowout, pipeline leak, and unknown facility with average emission rates of 31133 kg/h, 3990
kg/h and 8210 kg/h, respectively.

The LRAD algorithm utilized Sentinel-2 bands 3, 8, 11, and 12 to remove multi-type

artifacts associated with low reflectance in methane-sensitive bands, which greatly improved
feature extraction by the deep model especially in heterogeneous regions of O&G fields. We
applied the LRAD algorithm to ΔR retrieval from Sentinel-2 observations and compiled ΔR



datasets (1627 images in total) that include six different O&G super-emitters. The six labelled
datasets have various ratios of positive (plume-containing) to negative (plume-free) sample size,
plume sizes, and background noises.
The DSAN model was used to detect methane point sources based on ΔR images, aiming
to resolving challenges arising from the domain shift between Sentinel-2 ΔR images for
methane sources in different regions. For transfer detection tasks across six known methane
sources, the DSAN model achieved an average macro-F1 score of 0.86, outperforming
MethaNet and ResNet-50. Without a need for a huge volume of training data, our DSAN model
operated in a data-efficient manner which leveraged knowledge acquired from a source domain
during the training process to perform plume classification in a target domain.
Moving forward, the developed workflow can be modified to detect methane from other
multispectral instruments, including Sentinel-2, LandSat-8, and WorldView-3. Also, it has the
potential for detecting plumes of other pollutants observable by satellites such as $NO_2$ or $CO_2$.
Moreover, while this study made efforts to develop a labelling decision rule, the confidence of
the labels determined by human analysts was difficult to quantify. To facilitate robust algorithm
development, we recommend the development of standards for plume identification and
construction of benchmark plume datasets for varied satellite instruments.

**Data availability**

The six compiled methane retrieval ΔR datasets will be made available through a public
repository upon publication [https://doi.org/10.57760/sciencedb.15792].

**CRediT author statement**



**Shutao Zhao:** Conceptualization, Methodology, Data curation, Software, Visualization,
Writing-Original draft preparation. **Yuzhong Zhang:** Conceptualization, Investigation,
Supervision, Validation, Writing-Original draft preparation, Funding. **Shuang Zhao:** Funding.
**Xinlu Wang:** Reviewing and Editing. **Daniel J. Varon:** Software, Reviewing and Editing.
**Declaration of Competing Interest**
The authors declare that they have no known competing financial interests or personal
relationships that could have appeared to influence the work reported in this paper.
**Acknowledgements**
This work was funded by the National Key Research and Development Program of China
(2022YFE0209100), the National Natural Science Foundation of China (42307129), and the
Zhejiang Provincial Natural Science Foundation (LZJMZ24D050005).

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
