# Peer review of "A Data-Efficient Deep Transfer Learning Framework for Methane Super-Emitter"

_EGUsphere, 2024_

## Author Comment (AC1)

**REPLY TO REVIEWERS**

Dear editor and reviewers:

Thanks for your time and comments for our manuscript entitled "A Data-Efficient Deep Transfer Learning Framework for Methane Super-Emitter Detection in Oil and Gas Fields Using Sentinel-2 Satellite". The manuscript has certainly benefited from these insightful revision suggestions. Below we provide point-wise response to reviewers' comments. In the manuscript, revised or newly added sentences are in deep blue which relates to the reviewers' comments.

**[Major point 1]**

**The dataset description needs to be improved. As it is now, the description of various parts of the used data is split into many sections. Currently, the description of the training dataset starts in section 2.1, later the validation subset is detailed in 2.5.1 and the test initially mentioned in 2.5.2, while in more detail explained in section 3. Overall this makes it quite difficult to read, which data was used when. -> Please unify this and clear up the existing split sections.**

Reply:

Thank you for your valuable comment. We have reorganized Section 3 to consolidate all information related to the dataset into a single section, which should improve readability. As part of this, we moved the content from the original Section 2.5 into Section 3. Only the introduction to Sentinel-2 data has been retained in Section 2.1 (Satellite data). The revised structure of Section 3 is as follows:

3 Methane dataset and Experimental design

3.1 Methane retrieval ($\Delta R$) imagery dataset construction

3.1.1 Data preprocessing

3.1.2 Data annotation

3.2 Experimental design

3.2.1 Performance evaluation on transfer tasks

3.2.2 Real-world application for new source discovery

Please refer to manuscript Lines 264-384 for the specific changes.

**Additionally, some details are missing - practical ones which would make it easier to understand the exact training process used. For example, it is not exactly clear which input resolution does the data have - is it 200x200 px, or 224x224 px? - Please clarify this and explicitly write this in the paper. Please also detail the exact number of tiles used in**

**the different dataset splits.**

Reply:

Thank you for pointing this out. Following the suggestion, we have added the following information:

- For input resolution, we now specify that the Sentinel-2 observations were divided into 16 km² patches (200×200 pixels), which resulted in 200×200 ΔR matrix. We then applied a colormap to the matrix and resized them into 224×224×3 ΔR RGB images to align with the input of ResNet-50 structure and ensure compatibility with ImageNet-based pre-trained parameters. Please refer to manuscript Lines 306-310 for this clarification.

- Regarding the exact number of tiles used in the different dataset splits, for each '5→1' task, we roughly have 250 ΔR images in the source domain and other 250 in the target domain. Each '5→1' task contains about 1,250 ΔR images in the source domain and 250 images in the target domain. Please refer to manuscript Lines 348-350 and 351-342 for the specific details.

**Also please highlight if and what measures to detect overlap between samples in the created train/val/test datasets were used (temporal split? spatial split?). This is quite clear for the train/test sets, but less so for the validation set used for the other model architectures. A more explicit description of these details is expected in typical machine learning literature (which is not just reusing existing benchmark datasets).**

Reply:

Thank you for your constructive comment. To address your concern, we have supplemented explanation in Section 3.2.1 and Section 3.2.2, as well as Table S1. In this study, we employ two types of models: conventional CNN models like MethaNet, which involve train/val/test datasets, while DSAN does not require a validation set. Therefore, we deploy different partitioning ways of the training, validation, and test set for different models and tasks, as shown in Table S1. For conventional CNN models, which lack a feature adaptation layer, they directly rely on the feature mappings learned during the training process. So, an independent validation set is required to tune hyperparameters of the model and prevent overfitting. Under the '1→1' task, the training and validation sets are derived from one of the ΔR datasets listed in Table 3, with an 80:20 random split, and test sets are the target of transfer tasks (another ΔR dataset). Similarly, for the '5→1' task, five of the ΔR datasets are combined and shuffled, with an 80:20 random split for the training and validation sets, while the remaining dataset is used as the test set. DSAN is capable of adapting to distribution changes during training, eliminating the need for a validation set to tune hyperparameters. Under the '1→1' task, the training set (source domain) consists of one ΔR dataset, and the test set is another dataset from a different region. In the '5→1' task, the training set (source domain) includes five ΔR datasets, and the

test set is one from a different region.

**Table S1** Training, validation, and test sets separation for different types of models and tasks.

| Model | Task | Training set | Validation set | Test set |
|-------|------|-------------|----------------|----------|
| MethaNet, | '1→1' | $80\%Dataset\#x_i$ [a] | $20\%Dataset\#x_i$ | $Dataset\#x_j$ [b] |
| ResNet-50 | '5→1' | $80\%\sum_{i=1}^{n=5}Dataset\#x_i$ | $20\%\sum_{i=1}^{n=5}Dataset\#x_i$ | $Dataset\#x_j$ |
| | '1→1', | $Dataset\#x_i$ | *n/a* | $Dataset\#x_j$ |
| DSAN | '5→1' | $\sum_{i=1}^{n=5}Dataset\#x_i$ | *n/a* | $Dataset\#x_j$ |
| | application for new source detection | $\sum_{i=1}^{n=6}Dataset\#x_i$ | *n/a* | *3537 $\triangle R$ images* |

[a-b] $Dataset\#x_i$ and $Dataset\#x_j$ refer to one of the Dataset#1~#6 as listed in Table 3. Here, $i \neq j$, meaning that the source and target datasets in each task are distinct, ensuring no overlap between them.

**[Major point 2]**

**Right now, it is not clear how much have the proposed steps helped for the real data prediction. There, we see only one result, but it would be informative to see results from other used models. If it's possible and feasible, please add an ablation study of what would happen if we didn't use the proposed pre-filtering step LRAD? Would the scores degrade significantly? These results don't need to be analysed in such a detail as the rest, a simple single row of results in Table 4 would suffice (as the labeling has been already done for this data, it should be easy to recalulate these scores for another model variants).**

Reply:

Thank you for your constructive comment. We agree that it is important to assess how much the proposed steps have helped in methane plume detection on real data. To address this question, we conducted an ablation study on both the '1→1' and '5→1' transfer tasks, as well as the application experiment. The results indicate that LRAD shows little to no improvement on homogeneous surfaces, including Dataset#1, Dataset#4, Dataset#5, and application regions which feature homogeneous surface conditions. In contrast, it demonstrates significant improvement in heterogeneous regions, such as Dataset#2, Dataset#3, and Dataset#6. The LRAD denoising step significantly reduces the false negative rate but has little improvement on the false positive rate. We have summarized these results in Sections 6.1 "Impact of the denoising method on methane detection".

Please refer to manuscript Lines 515-545 for the specific details.

Lines 525-545:

To further investigate the impact of LRAD, we conduct additional ablation experiments

using the 30 '1→1' tasks and 6 '5→1' tasks (Section 2.5.1) to evaluate its influence on model performance. Fig. 14 shows the macro-F1 scores for the transfer tasks using the detection model without applying LRAD (WoLRAD-DSAN). The average macro-F1 scores are 0.72 (ranging from 0.53 to 0.92) for the '1→1' tasks and 0.74 (ranging from 0.55 to 0.92) for the '5→1' tasks, significantly lower than those achieved by the LRAD-DSAN model on the same tasks (Fig. 8). The effect of LRAD is particularly pronounced on Dataset #2, #3, and #6, where the absence of LRAD leads to macro-F1 score reductions of approximately 33%, 28%, and 23%, respectively. These results from the ablation experiments are consistent with the SNR improvements shown in Fig. S7, further underscoring the efficacy of LRAD, especially over heterogeneous surfaces.

The impact of LRAD can be further elucidated by analyzing the false negative rate (FNR) and false positive rate (FPR) for the transfer tasks (Fig. S10). LRAD substantially reduces the FNR (e.g., by 52%, 46%, and 33% for Dataset #2, #3, and #6, respectively), but only moderately reduces the FPR by 3% to 24%. These findings demonstrate that LRAD is primarily effective in reducing omissions of methane super-emitters.

[Figure]

**Fig. 14.** Macro-F1 scores on the transfer tasks given by the WoLRAD-DSAN model. Each square represents a transfer task. '5→1' indicates that the source domain is fused from five datasets excluding the target domain dataset. WoLRAD-DSAN refers to the DSAN framework without incorporating the LRAD denoising algorithm.

**[Minor point 1]**

**The background literature sections is missing some recent works. For example [Růžička, V., Mateo-Garcia, G., Gómez-Chova, L. et al. Semantic segmentation of methane plumes with hyperspectral machine learning models. Sci Rep 13, 19999 (2023).**

**https://doi.org/10.1038/s41598-023-44918-6] proposed a U-Net based model working with a mixture of source instruments and shows performance for both multispectral (WorldView-2) and hyperspectral data (AVIRIS-NG and EMIT). Relevant to this work, it also demonstrated zero-shot generalisation. That is using a model pre-trained on first dataset (with relatively local samples) on data from near-global sensor (with larger diversity of background scenes) -> please relate this to the used source / target domain adaptation used in this work. It also should be added to page 4 among other deep learning techniques used to detect methane leaks in multispectral and hyperspectral data.**

Reply:

Thank you for pointing out this important relevant work. The study by Růžička et al. (2023) is indeed closely related to our study, as both approaches use data from labeled source domain to build models for tasks in the unlabeled target domain, addressing the challenge of limited training data. We have added a reference to this work in Line 99 and discussed its relevance to our study, particularly in the context of zero-shot generalization and deep transfer learning, in Lines 114-123.

**[Minor point 2]**

**Please describe which exact model variant was used for the real data prediction (results shown on Fig 11) - can this be related to one of the already used scenarios (1-1, 5-1, ...)?**

Reply:

Thank you for your comment. We have supplemented a description of the exact model variant used for the real data prediction in Section 3.2. Specifically, we applied DSAN to detect new sources (target domain), where the training set (source domain) consists of Dataset#1–#6 (1627 ΔR images), and the test set (target domain) includes 3527 ΔR images collected from the application area. Please refer to Lines 381-383 and Table S1 for further details.

**[Minor point 3]**

**On page 8 clarify if the used formula is the multiband-multi-pass (MBMP) method of [Varon, D. J., et al. High-frequency monitoring of anomalous methane point sources with multispectral Sentinel-2 satellite observations, Atmos. Meas. Tech., 14, 2771–2785, https://doi.org/10.5194/amt-14-2771-2021, 2021.] and name it as such, or highlight if there are any notable differences.**

Reply:

Thank you for your suggestion. To clarify, we did not use the MBMP method, but instead

employed the band ratio method, as used in Ehret et al. (2022) and Irakulis-Loitxate et al. (2022), which directly calculates band differences. We have clarified this in Line 164 of the manuscript by explicitly stating the method used in the calculation.